# Generalized Analytical Model for Enzymatic BioFET Transistors

**DOI:** 10.3390/bios12070474

**Published:** 2022-06-30

**Authors:** Cristian Ravariu, Avireni Srinivasulu, Dan Eduard Mihaiescu, Sarada Musala

**Affiliations:** 1BioNEC Group, Department of Electronic Devices Circuits and Architectures, Faculty of Electronics ETTI, Polytechnic University of Bucharest, Splaiul Independentei 313, 060042 Bucharest, Romania; 2School of Engineering & Technology, K.R. Mangalam University, Gurugram 122103, Haryana, India; avireni@ieee.org; 3Faculty of Engineering, JECRC University, Jaipur 303905, Rajasthan, India; 4Organic Chemistry Department, Faculty of Chemistry, Polytechnic University of Bucharest, Splaiul Independentei 313, 060042 Bucharest, Romania; dan.mihaiescu@upb.ro; 5Department of Electronics and Communication Engineering, Vignan’s Foundation for Science, Technology and Research, Guntur 522213, Andhra Pradesh, India; sarada.marasu@gmail.com

**Keywords:** ENFET modeling, enzyme kinetics, bio-nano-electronics

## Abstract

Software tools that are able to simulate the functionality or interactions of an enzyme biosensor with Metal Oxide Semiconductor (MOS), or any Field Effect Transistor (FET) as transducer, represent a gap in the market. Bio-devices, or Enzyme-FET, cannot be simulated by Atlas or equivalent software. This paper resolves this issue for the enzymatic block coupled with FETs’ role within biosensors. The first block has the concentration of biological analyte as the input signal and concentration of ions from the enzymatic reaction as the output signal. The modeling begins from the Michaelis–Menten formalism and analyzes the time dependence of the product concentrations that become the input signal for the next FET block. Comparisons within experimental data are provided. The analytical model proposed in this paper represents a general analytical tool in the design stage for enzymatic transistors used in clinical practices.

## 1. Introduction

Enzyme biosensors have significantly developed within the last 20 years [1,2,3,4], with modern trends indicating a down-scaling in biosensor sizes [4]. The proposed solution for the next 10 years is the co-integration of the enzyme receptor with Metal Oxide Semiconductor (MOS), or any Field Effect Transistor (FET) as transducer, to produce an ENFET (Enzyme-Field Effect Transistor) [5]. Models of enzymatic kinetics have been developed in the field of biochemistry; however, software tools that are able to simulate the functionality and technology of an ENFET are lacking. Silvaco’s platform has resources to simulate image sensors, magnetic, or optical sensors [6], but it does not have facilities to simulate an ENFET.

The lack of models for integrated biosensors is critical in the design stage, though some primary models have recently been developed for glucose biosensors [7,8]. This paper generalizes the previous biosensor concept and extends the glucose-oxidase receptor [8] to any enzymatic layer as well as the reaction rate from 1-order to 2-order and to n-order, n > 1. BioFET transistors consist of biosensors composed of a general biological receptor coupled with a FET transducer [9,10,11]. If the receptor consists of an Ion Sensitive Electrode (ISE), then the BioFET is of the ISFET type [11]; if the biological receptor consists of an enzymatic membrane, then the BioFET is of the ENFET type [7,8]. Recently, for Organic Electro-Chemical Transistors (OECT) biosensors, the change in product concentration was translated into a shift of the threshold voltage [12]; a similar theory was proposed for ISFET biosensors [11]. The ISE components from ISFETs are largely modeled by electrochemistry. In this paper, Nerst’s equation is also used to express the dependence of the gate potential versus the ions concentration. The gate voltage of an ENFET has two components: (i) the previous discussed Nerst potential, and (ii) a voltage from an external power supply connected through an integrated reference electrode. Due to the technological compatibility with Si wafers, the Ag/AgCl electrode is often used with Silicon ENFETs [9]. For the modeling of the FET component, we assume that the accumulated ions from the enzymatic reaction will produce a deviation of the threshold voltage, which will influence the drain current of the transistor. The main blocks of an ENFET biosensors are represented in Figure 1.

This paper is focused on the enzymatic block modeling, which is identified as the missing part in Silvaco tools. General notations from this paper are:

[P]—the Product concentration in [mol/L]; k—the global reaction rate constant; K—the aspect factor of the transistor [μA/V^2^]; [A]—analyte concentration [mol/L]; v_max_—maximum reaction rate [mol/L·s]; K_M_—Michaelis constant [mol/L]; t—measurement time [s]; E—enzyme loading [mol/L]; ε_0_—vacuum dielectric permittivity; R—the gases constant; F—Faraday’s constant; N_A_—Avogadro’s number; T—temperature [K]; n—the reaction order; and e—elementary electric charge.

## 2. The ENFET Modeling

### 2.1. Enzyme Block Modeling

The enzyme receptor block modeling starts from the equations given in [13]. From there, we discover the time dependence of the analyte/product concentrations, which represents the input/output signals of the enzyme block from Figure 1.

The reaction rate, v, depends on instantaneous value of the concentration of the consumed analyte, or concentration of the formed product:(1)$$v=-\frac{d[A]}{dt}=\frac{d[P]}{dt}$$

The product concentration [P], after a measurement time t, is given by:(2)$$[P](t)=\int_{0}^{t}v(t)dt$$

In the actual modeling, we consider the following initial conditions: [P(t = 0)] = 0, signifying that no product exists before enzyme reaction and [A](t = 0)] = A_0_ for the initial analyte concentration.

For an n-order reaction, n > 1, the reaction rate depends on the analyte concentration to power n:(3)$$v=k\cdot {\left[A\right]}^{n}$$

Combining the first equality from (1) and (3), we obtain:(4)$$k\cdot {\left[A\right]}^{n}=-\frac{d[A]}{dt}$$

After the variables’ separation, we obtain:(5)$$-kdt=d[A]/{\left[A\right]}^{n}$$

Here, we integrate the first member after t, while in the second member we integrate it after [A]:(6)$$-\int_{}^{}kdt=\int_{}^{}\frac{d[A]}{{\left[A\right]}^{n}}$$

The result of the integral offers the analyte concentration in time:(7)$$\frac{{\left[A\right]}^{-n+1}}{-n+1}=-kt+ct$$
where ct is an integration constant. It is extracted from the initial condition: [A(t = 0)] = A_0_. Replacing this constant in (7), results the analyte concentration:(8)$$[A](t)={\left[A_{0}{}^{1-n}+(n-1)\cdot k\cdot t\right]}^{\frac{1}{-n+1}}$$

From there, the reaction rate versus time is $v=k\cdot {\left[A\right]}^{n}$:(9)$$v(t)=k\cdot {\left[A_{0}{}^{1-n}+(n-1)\cdot k\cdot t\right]}^{\frac{n}{-n+1}}$$

The product concentration can be estimated by Equations (2) and (9):(10)$$[P(t)]=ct^{\prime }-{\left(A_{0}{}^{1-n}+(n-1)\cdot k\cdot t\right)}^{1/(-n+1)}$$
where the constant of integration ct′ is a result of the initial condition, [P](t = 0) = 0, in (10). The final result is:(11)$$[P](t)=A_{0}-{\left[A_{0}{}^{1-n}+(n-1)\cdot k\cdot t\right]}^{\frac{1}{1-n}}$$

The previous algorithm will be applied for each particular case of reaction in order of 0, 1, and 2, respectively. After the final evaluation of the integration constants, the results following the time dependences of the analyte concentration [A], reaction rate, v, and product concentration [P] will be clarified.

For zero-order reactions:(12)$$[A](t)=A_{0}-kt$$
(13)v(t)=k
(14)$$\left[P](t)=kt=A_{0}-[A](t\right)$$

For first-order reactions:(15)$$\left[A](t)=A_{0}\cdot \exp(-k\cdot t\right)$$
(16)$$v(t)=k\cdot A_{0}\cdot \exp(-k\cdot t)$$
(17)$$[P](t)=A_{0}\cdot \left(1-\exp(-k\cdot t)\right)$$

For second-order reactions:(18)$$[A](t)=\frac{1}{A_{0}^{-1}+k\cdot t}$$
(19)$$v(t)=\frac{k}{{\left(A_{0}^{-1}+k\cdot t\right)}^{2}}$$
(20)$$[P](t)=A_{0}-\frac{1}{A_{0}^{-1}+k\cdot t}$$

The proposed model from this paper is valid for enzymes that have a single substrate. A solution for enzymes that have multiple substrates must estimate the global reaction rate if the substrates are to be useful analytes. This procedure is known as the initial rate method [13]. From there, only one reactant concentration is varied, while all the others are kept constant. The order of that particular reactant is calculated while measuring the rate at the beginning of the reaction as instantaneous rate extrapolated to time zero. The same procedure is repeated with all other reactants. The global reaction order is given by the superposition of the individual orders if there are multiple substrates. This procedure will need an initial experimental study of enzymes within all substrates. The existence of multiple substrates means there is only one analyte of interest, while the others are secondary, ineffectual substrates; the action of the secondary substrates can be alleviated by inhibitors [14].

### 2.2. FET Part Modeling

The FET transducer can be a typical MOSFET [9], a pseudo-MOS [15], an OECT [12], or any FET, including a Carbon-nanotube CNT-FET [16] or nano-wire-FET [17] because the modeling for the aforementioned FETs is based on similar Spice models [18,19,20]. The organic thin film transistors (OTFT) [21], sometimes applied in biosensors [12], obey to the same formalism, which produces the specific significance of their model parameters [22]. If the transistor is operated in a saturation regime, the drain current of any FET is dominated by the general Spice model [18]:(21)$$I_{D}=K/2\cdot {\left(V_{GS}-V_{T}\right)}^{2}$$
where I_D_ is the drain current, K is the aspect factor, V_GS_ is the gate-source voltage, and V_T_ is the threshold voltage of the FET part. The aspect factor is [18]:(22)$$K=W/L\cdot \mu_{c}\cdot C_{tot}^{}$$
where W is the width, L is the length of the channel of the transistor, μ_c_ is the carrier’s mobility, and C_tot_ is the total gate capacitance. The total gate capacitance is composed of a series of capacitances of each insulator over the Si-wafer, (e.g., SiO_2_, HfO_2_, Si_3_N_4_, TiO_2_, etc.), including the enzymatic layer, which is an aqueous environment during the analyte tests:(23)$$C_{tot}^{-1}=\sum_{i}\frac{t_{i}}{\varepsilon_{i}}$$
where i is the index of each insulator, t_i_ is the thickness of the insulator “i” given by technology, and ε_i_ is the dielectric permittivity of the insulator “i”. The gate-source voltage is composed by a value, V_ext_, externally applied from a DC power supply in respect to a reference electrode (RE, in Figure 2) and the Nerst potential V_N_ of the gate electrode in contact with the tested liquid with ions:(24)$$V_{GS}=V_{\mathrm{ext}}+V_{N}$$

The Nerst potential logarithmically depends on the concentration of the ionic products:(25)$$V_{N}=V_{0}+RT/zF\cdot \ln[p]$$
where V_0_ is the standard electrode potential, z is the valence number, p is the ions concentration expressed in normed units to mol/L, and RT/F is the thermal voltage that is 0.025 V at room temperature.

The threshold voltage is a superposition between the threshold voltage of the ENFET structure in absence of any analyte, V_T,FET_, and the shift of the threshold voltage, ∆V_T_, due to the ions formation in the tested solution:(26)$$V_{T}=V_{T,FET}+\Delta V_{T}$$

After the tested solution is dropped on the enzymatic membrane and after measuring time t, the enzyme block produces ions, in concentration [P] as its output signal. These products of the enzymatic reaction are usually ions, such as H^+^ or O^−^ (the [P] concentration of which has been modeled in the Section 2.1). The concentration [P] that is expressed in moles/liter must be transformed into the surface charge Q_c_ expressed in electrons per cm^2^, because the formula for calculating the threshold voltage deviation is [19,23]:(27)$$\Delta V_{T}=\frac{Q_{c}}{C_{H}}$$
where C_H_ is the specific capacitance of the Helmholtz layer, expressed in F/cm^2^. In this paper, we assume that the ionic charge is mainly distributed according to the Helmholtz model of the rigid layer; hence, the surface charge Q_c_ is placed at a distance t_H_ from the surface of the working electrode, where t_H_ is the thickness of the Helmholtz layer (Figure 2). Other authors also sustain that the enzymatic layer behaves as a membrane situated near the outer Helmholtz plane of the gate electrode [24].

Despite any ions in deep diffusion layers, by applying the average theorem of the integral, the diffusion charge can be equivalent to an additional component of the surface charge also expressed in electrons per cm^2^ [19]. Consequently, the actual model does not suffer from the equations point of view, but only Q_c_ charge can change its value. In the future, improvements can be made by adjusting the Q_c_ value with the modeling of these diffused ion tails.

The surface charge Q_c_ is computed step by step: (i) the volume ions density results from the ions product concentration [P] as ([P] · N_A_) expressed as ions number/cm^3^; (ii) the surface ions density results from the number of ions per cm^3^ multiplied by the thickness of the Helmholtz layer that means ([P] · N_A_ · t_H_) expressed in number of ions per cm^2^; (iii) the surface electric charge Q_c_ results from the number of ions per cm^2^ multiplied by the charge of one ion, including its ± sign (e.g., ±z is replaced by −2 for O^−^):(28)$$Q_{c}=\pm z\cdot e\cdot [P]\cdot N_{A}\cdot t_{H}$$
where the notations were previously explicated. The specific capacitance of the Helmholtz layer depends on the dielectric constant, ε_0_:(29)$$C_{H}=\varepsilon_{0}/t_{H}$$

Replacing (28) and (29) in (27) we obtain:(30)$$\Delta V_{T}=\frac{\pm z\cdot e\cdot [P]\cdot N_{A}\cdot t_{H}^{2}}{\varepsilon_{0}}$$

Replacing (30) in (26) and adding (25), (24), (23), and (22) in (21), results in the drain current depend on the product concentration:(31)$$I_{D}=\frac{W}{L}\cdot \frac{\mu_{c}}{2}\cdot {\left(\sum_{i}\frac{t_{i}}{\varepsilon_{i}}\right)}^{-1}\cdot {\left(V_{\mathrm{ext}}+V_{0}+RT/zF\cdot \ln[p]-V_{T,FET}\pm \frac{z\cdot e\cdot [P]\cdot N_{A}\cdot t_{H}^{2}}{\varepsilon_{0}}\right)}^{2}$$

This is the transduction curve of the FET part from Figure 1.

## 3. Parameters Extraction

### 3.1. Parameters Extraction for the Enzy Block

Within the Michaelis–Menten formalism, the initial reaction rate, v, is dependent on the analyte concentration, [A]:(32)$$v=\frac{v_{\max}\cdot \;[A]}{K_{M}+[A]}$$

The model parameters are the Michaelis constant K_M_ and the maximum reaction rate, v_max_. The 1/v_max_ value can be estimated by a Lineweaver–Burk plot, where the line among measured points −1/v versus 1/[A] intercepts the vertical axis; similarly, K_M_ is extracted. A generalization of model (32) is the Hill equation:(33)$$v=\frac{v_{\max}\cdot {\;[A]}^{h}}{K_{M}^{h}{+[A]}^{h}}$$
where h is the Hill coefficient, K_M_ is analogous to the Michaelis–Menten constant, representing the value of analyte concentration at which v = v_max_/2. This additional parameter is h > 1 for positive cooperativity or 0 < h < 1 for negative cooperativity. In biosensors, enzymes with positive cooperativity allow exceptional sensitivity of analyte concentrations.

For all models of [P], the only parameter left to be extracted is k, the reaction constant. If the analyte concentration is high, (e.g., [A] >> K_M_) in relations (32), (33), then these relations can be approximated at [A] → ∞ by:(34)$${v=v}_{\max}\cdot {\;[A]}^{0}$$
so that k can be approximated to v_max_ for zero-order reactions.

If the analyte concentration is extremely low (e.g., [A] << K_M_) in relation (32), then this relation can be approximated with:(35)$$v=\frac{v_{\max}}{K_{M}}\;\cdot {\left[A\right]}^{1}$$
so that k can be approximated to v_max_/K_M_ for first-order reactions.

If the analyte concentration is very low (e.g., [A] << K_M_) in relation (33), then this relation can be approximated with:(36)$$v=\frac{v_{\max}}{K_{M}^{h}}\cdot {\;[A]}^{h}$$
so that k can be approximated to v_max_/(K_M_)^h^ for h-order reactions in the instance that the h parameter matches with the reaction order.

If previous methods are not available, another solution is to estimate k_2_ using the enzyme loading E expressed in mol/L and v_max_ value [13]:(37)$$k_{2}=v_{\max}/E$$

Parameters for the extraction for some experimental enzymatic biosensors are approached in Section 4.

### 3.2. Parameters Estimation for the FET Part in a Design Stage

Previous analytical models become useful tools if they are applied in the design stage of an ENFET. For example, designing a glucose biosensor as an ENFET requires measuring the O^−^ ion products concentration after the enzymatic reaction in the range 1 mM–1000 mM, and selecting a MOS-FET transducer, 350 nm technological node [25]. For this technology we can use [25]: W = 5 mm, L = 0.35 mm, V_T, FET_ = 1.5 V, t_I_ = 40 nm and I is only SiO_2_. Other values can be estimated from the solid-state physics: μ_c_ = 800 cm^2^/Vs for electrons in channel, V_0_ = +0.79 V for Ag gate electrode, t_H_ = 100 nm for micro-molar range of ions concentration [24], dielectric permittivity of the Helmholtz layer is approximated to that of a vacuum. The thickness of the enzymatic region is given by the immobilization technology, offering possible values of t_E_ = 1 mm [7,8,9]. Relative dielectric permittivity of the enzymatic membrane is approximated to that of water ε_r,Enzy_ ≅ 80.

The total gate capacitance can be computed by Equation (23): C_tot_ ≅ 0.7 × 10^−10^ F/cm^2^. The aspect factor K is given by Equation (22): K = 0.8 μA/V^2^. The shift in the threshold voltage is computed by Equation (25) starting from 0.109 V at [O^−^] = 1 μM, up to 1.14 V at [O^−^] = 1 mM.

For the Ag gate electrode, according to Equation (25), the Nerst potential varies from 0.876 V at [O^−^] = 1 μM, till 0.790 V at [O^−^] = 1 mM.

The external voltage V_ext_ must be selected so that the transistor always works in strong inversion, meaning that V_GS_ − V_T_ > 0. In the case of this study, we selected V_ext_ = 2 V. Replacing these values in the global Equation (31), we obtain a drain current excursion from 0.882 μA at [O^−^] = 1 mM to 2.361 μA at [O^−^] = 1 mM.

In the next paragraph, the previous model is compared to select experimental biosensors.

## 4. Results of the Analytical Models Applied to Enzymatic Biosensors

### 4.1. Results for a BioFET with Glucose-Oxidase

Upon initial analysis, a glucose biosensor based on Glucose-Oxidase (GOX) enzyme receptor is theorized. The Michaelis constant K_M_ = 10 mM and v_max_ = 0.018 mol·L^−1^·s^−1^, parameters were collected from the literature [13].

The fastest reaction has 0-order with a time response of 29 s, followed by 32 s for 1-order, 45 s for 2-order, and 292 s for 3-order, declaring that [P] reaches 99% from [A](t = 0) = 5 mM in Figure 3. The effect of the Hill parameter h inside Equation (33) is considered in Figure 4. The glucose (analyte) concentration ranges between 0.01 and 10 mM. The maximum velocity v_max_ = 1.74 mmol/L.s is estimated by a Lineweaver–Burk plot from real data [9,13].

The graphical aspect is similar to the experimental test [13].

The Michaelis constant can take different values versus the usual value of 10 mM [13] in respect to the technological process: K_M_ = 13.55 μM when GOX is immobilized on sol-gel derived nanostructured CeO_2_ film, K_M_ = 2.9 mM when GOX is immobilized on ZnO nano-rods, and K_M_ = 6.08 mM when GOX is immobilized on porous nano-crystalline TiO_2_ [26]. Simulations from Figure 5 of the analytical models are in agreement with the experimental results [12].

### 4.2. Results for a BioFET with Acetyl-Cholinesterase

A second example concerns an acetyl-cholinesterase (AcHE) biosensor based on pH-Field-Effect Transistor that detects acetylcholine (AcH) and measures the Hydrogen ions according to the reaction:$${\mathrm{AcH}+H}_{2}O\;\overset{\mathrm{AcHE}}{\to }{\mathrm{Choline}+\mathrm{CH}3\mathrm{COO}}^{-}{+H}^{+}$$

A fabricated AcH biosensor uses an enzyme loading of 2 × 10^−5^ mol/L AcHE with the analyte concentration of 4 × 10^−3^ mol/L AcH [13]. The AcHE has activity of 518 units/mg and molar mass of 280 kDa. In Figure 6, the experimental points were collected from [27], for a time interval of 0–240 s.

The reaction rate constant ranges from 10 mM^−^^1^s^−^^1^ to 10^6^ mM^−^^1^s^−^^1^ in experiment [27]. The analytical model (20) is applied to simulate the product concentration in Figure 6.

To validate our modeling, the analytical model is compared to some experimental picked points reported in the literature. Figure 7 presents the time dependence of the ionic products concentration of a biosensor with the AcHE enzyme for detection of the AcH analyte, using analytical model (17) and picked points for an experimental biosensor [27].

In [27], experimental curves were selected that correspond to k_s’1_ = 500 s^−1^ for Experim. ks’1 and k_s’2_ = 5 s^−1^ for Experim. ks’_2_. The simulated curves given by Equation (17) are calibrated to few experimental points [27], changing the reaction rate constant, while the experiment imposes the values of the initial analyte concentration, [A](t = 0) = 4 × 10^−3^ M, enzyme loading E = 2 × 10^−5^ M, and time framing - useful data for Equation (37).

## 5. Discussions

In other experiments, the entire transfer function of a biosensor is demanded. An experimental dependence of the biosensor current versus the analyte concentration is picked-up from the literature [28]. We further propose a plan to find the characteristic parameters of an equivalent ENFET transistor, which will obtain the same concentration-current curves as the enzymatic biosensor described in [28].

The ENFET transistor design algorithm is as follows:-We begin with the interval of the analyte concentration; [A] varies between 0.1 mM and 12 mM in this case.-Once we know the enzymatic reaction, the order of the reaction, and the extracted values of v_max_ and K_M_ from kinetics studies, the reaction constant is estimated by the models (34)–(37). In this case, the equivalent ENFET will also use tyrosinase enzyme to detect bisphenol A, as the experimental biosensor [28]. From the measured Lineweaver–Burk plots [28], v_max_ = 0.055 mM/ls and K_M_ = 1.13 mM are estimated. For the curve “Experim.1” from Figure 8, the analyte concentration is higher than K_M_, so that k = v_max_ according to Equation (34); for the curve “Experim.2” from Figure 8, the analyte concentration is smaller than K_M_, so that k = v_max_/K_M_ according to Equation (35).-After finding the reaction rate and analyte concentration, the product concentration can be extracted. In this case, the Equations (14) and (17) are used for the curves “Experim. 1” and “Experim. 2”, respectively. The product concentration is available in Figure 8, on the right axis, for the actual designed ENFET.-Once the range for the product concentration has been discovered, the entire algorithm from the Section 3.2 can be applied.-Finally, Equation (31) offers the drain current that depends on all FET parameters. In this case, it is was represented for different K and V_ext_ values in Figure 8, (see left axis).

In this case, a more advanced MOS-FET technology is selected: L = 45 nm, V_T,FET_ = 0.35 V, the Ag gate electrode is compared with previous physical parameters, and W can be adapted to accomplish optimal K values. In Figure 8, the optimal curve needs K = 1.42 mA/V^2^ and V_ext_ = −0.11 V, which leaves a W/L ratio of 25. For the optimal curve, the drain current excursion is 1–450 nA. The benefits of working with advanced MOS nano-metric technology, like sub-45 nm, are: ultra-low sizes, low threshold voltages, sub-40 nm insulator thickness, low currents, low external voltage, and low power consumption.

## 6. Conclusions

This paper has reported some analytical models developed on two directions: (i) an enzymatic block modeling and (ii) shift in the threshold voltage with the income signal from the enzymatic block within biosensors with a FET transducer. The time dependence of the concentration of the reaction products depends on the measurement time and initial analyte concentration according to various equations for each order of reaction. In this scope, the enzymatic block design requires some parameters of extraction. These are the partial results of both the enzyme entrapping technology, such as enzyme loading, and separate kinetics studies, such as the Michaelis constant, maximum velocity, and reaction rate constant. Once the designer knows the concentration range of the reaction ions produced from the enzyme block, they can use this ion concentration as an input signal for the FET part of the ENFET. The proposed global model of the drain current of an FET transistor in saturation depends on the incident ion concentration and a set of specific model parameters. Comparisons of the proposed models with experimental curves from the literature were in agreement.

## Figures and Tables

**Figure 1 biosensors-12-00474-f001:**
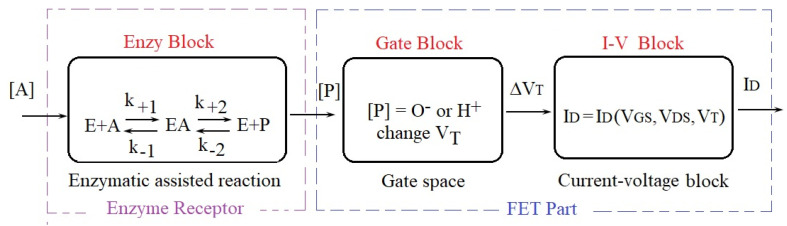
The blocks within a ENFET biosensor with enzyme receptors.

**Figure 2 biosensors-12-00474-f002:**
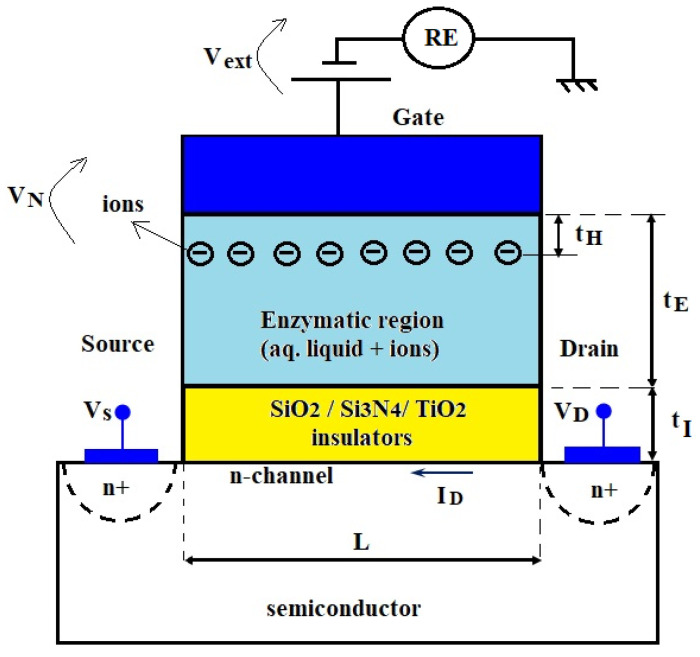
The structure of the modeled ENFET, where t_I_ is the sum of thicknesses for all inorganic insulators and t_E_ is the thickness of the enzymatic region; the n-type channel and the negative ions are considered as a possible example.

**Figure 3 biosensors-12-00474-f003:**
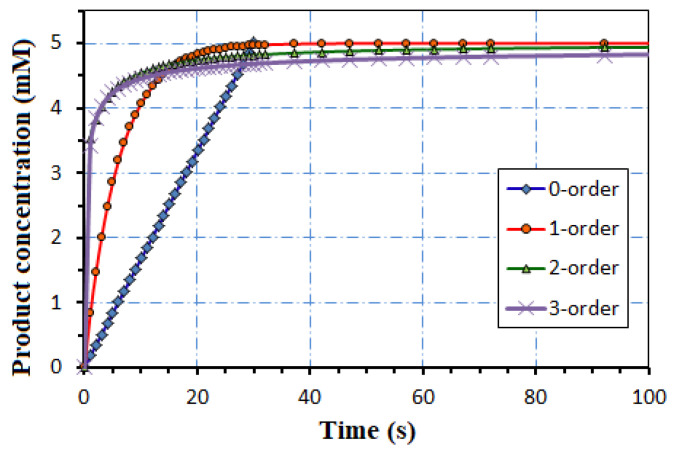
The time dependence of the Product concentration for reaction order n = 0, 1, 2, 3.

**Figure 4 biosensors-12-00474-f004:**
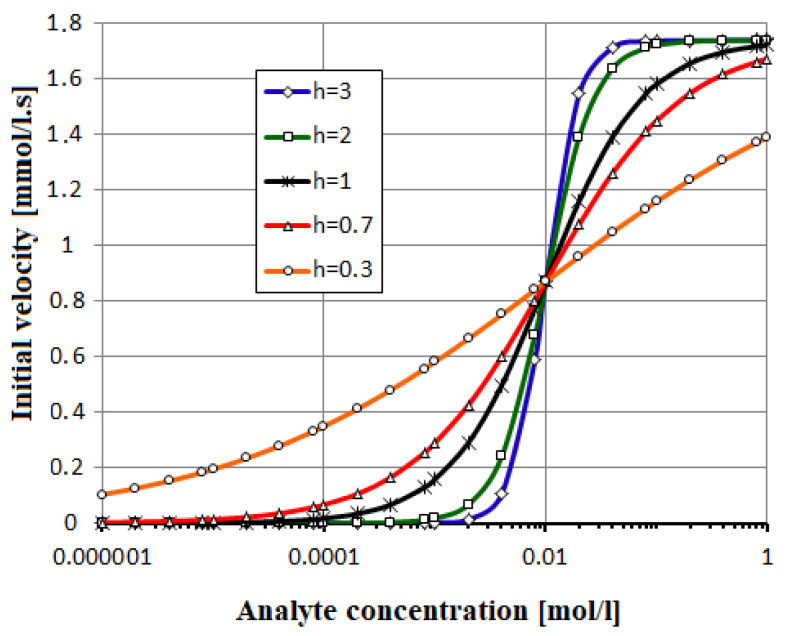
The enzymatic reaction is depicted by Hill model, with h = 0.3, 1, 2, 4.

**Figure 5 biosensors-12-00474-f005:**
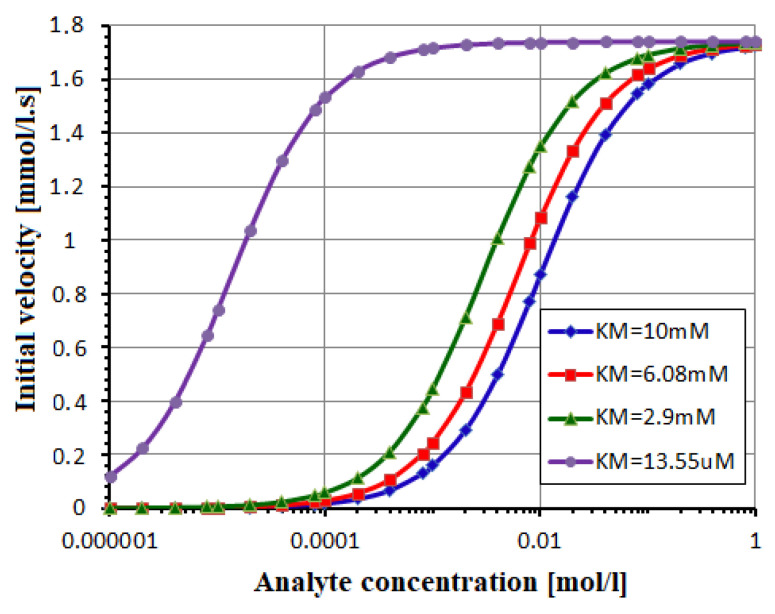
The v-[A] dependence from Equation (32) for different concentrations.

**Figure 6 biosensors-12-00474-f006:**
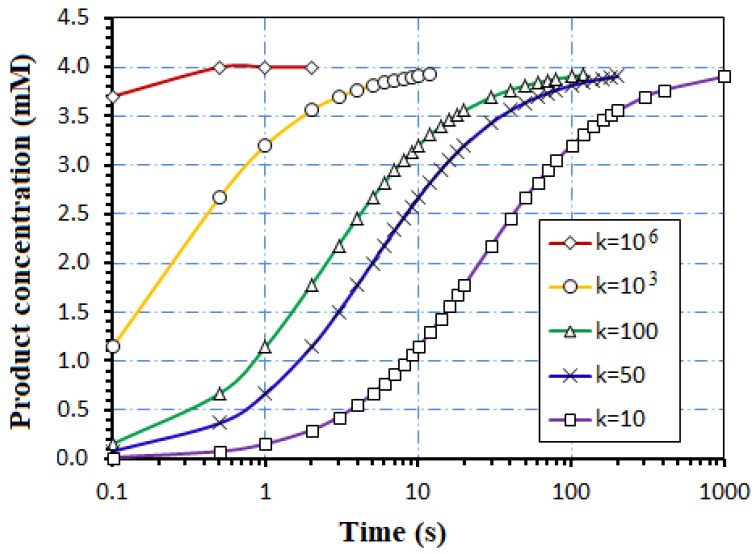
The simulated time dependence of the H^+^ product, for AcH biosensor, where k is the reaction constant expressed in mili-mol^−1^ s^−1^.

**Figure 7 biosensors-12-00474-f007:**
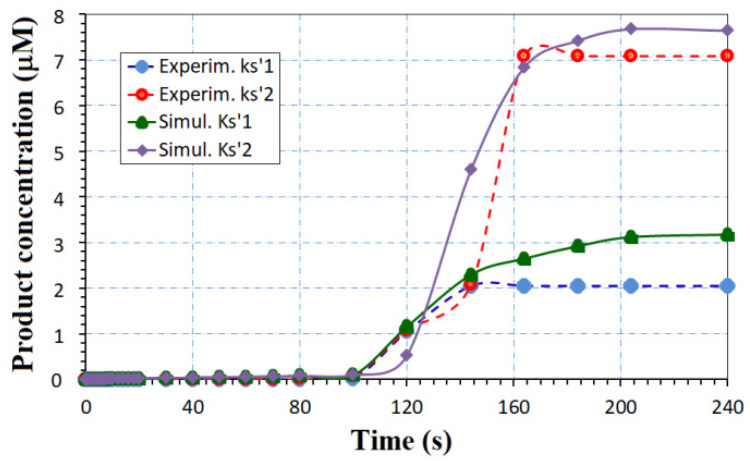
Comparison between the analytical model (17) and few experimental picked points. Experimental points were adapted with permission from Ref. [27].

**Figure 8 biosensors-12-00474-f008:**
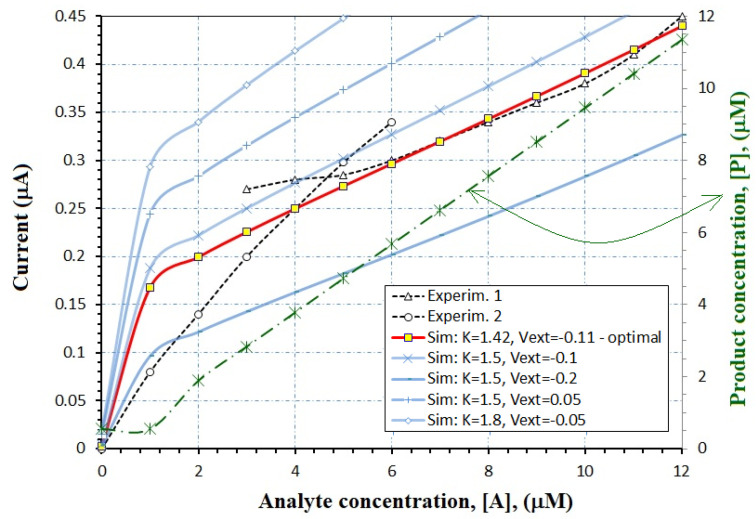
Analytical models are compared with experimental points from an enzymatic biosensor reported in [28]; the aspect factor K is expressed in mA/V^2^ and the voltage V_ext_ is expressed in V. Adapted from [28].

## Data Availability

Not applicable.

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
