# Peer review of "Generalized Analytical Model for Enzymatic BioFET Transistors"

_biosensors, 2022, doi:10.3390/bios12070474_

Round 1

Reviewer 1 Report

In this paper titled "Generalized Analytical Model for Enzymatic Bio-FET transistors", the authors try to describe a model to simulate the functionality or technological process of an enyzme biosensor with MOS or FET. However, the paper in the current form cannot be accepted for the following reasons:

1. The model established in this paper is way too simple. The authors only presented a model for the enyzme block while that for the FET block is still missing. In other word, the authors deducted a model for different orders of enzymatic reactions and obtained the time dependence of the product concentrations. In fact, this part has already been known for a long time. The key part for modeling a MOS or FET based biosensor is the relationship between the product concentrations and the final signal output, which is completely ignored in the work.

2. There are still some errors in the deduction, e.g. Eq. (3), Eq. (15) and Eq. (16). Besides, Eq. (4) and Eq. (5) are never used, what is the meaning for presenting these equations? Eq. (12) is obtained by combining Eq. (11) and Eq. (6) that is not explained clearly.

3. The time dependence of the product concentrations obtained in this paper is only valid for enzymes that have a single substrate. What about enzymes that have multiple substrates?

Author Response

I attached our reply in a Word file. 

Reviewer 2 Report

In this work the authors provide a generalized model for enzimatic bio-FETs. The development of this kind of model would indeed be beneficial to the FETs biosensors community, however in my opinion some major revision are necessary before the publication of this work.
1 - First of all, the authors describe in details the equation necessary to calculate the product concentration [P] in the Michealis-Menten formalism, but they do not provide an equation describing how [P] influences the characteristics of a transistor. The change in product concentration should be translated in a shift of the applied Vgs (or equivalently, in a shift of the threshold voltage), and included in the equations describing the FETs operation, similarly to what is commonly done for OECTs biosensors (https://doi.org/10.1038/s41598-020-70365-8).

2 - In Figure 6 the experimental results follow a sigmoidal trend, while the simulations seem to follow a linear trend. Why?

3 - In the last part of the discussion the authors find the optimal device paramters to fit the experimental data, but how do these parameters compare with the actual ones exploited in the experiment?

Moreover, please increse the resolution of Figure 1, it is very difficult to read.

Author Response

My reply is an attached Word file at this message. 

Reviewer 3 Report

Dear Authors,

Congratulations authors on your manuscript. Hope you can consider the below pointers to help better your manuscript.

1.     Take note of double spaces in the affliation and also manuscript. Change them to single spaces. 

2.     In the abstract, insert "that are" after "software tools". Consider changing "technological process" to "interactions"? Consider using simplier word other than "desideratum"? 

3.     Revise stentence line 20 for increased legibility.

4.     In line 37, consider changing "nowadays industry".

5.     In line 42, do explain the term BioFET. Consider explaining the "difficult task to detect complicate molecules from living matter is reduced to ions detection by ISE."

6.     In Figure 1, try not to use 3D. Increase font and colour legibility of text.

7.     Consider revising line 85 into formal english.

8.     Decrease font for super-scripts such as the power in formuala 14.

9.     In line 143, why was this leterature used? Can it be applied to another literature for better support?

10.  Caption for figure 4 should be "concentration" not "technologies"?

11.  Line 178, Why was inhibitor presence that was added for t > 4min ignored? Why 4 minutes?

12.  Line 181, check concentration superscripts

13.  Figure 5 caption, state what is k referring to.

14.  In conclusion, line 236, it should be "enzyme-assisted"?

15.  In author contribution, elaborate on "other FETs than MOS"?

I noticed that your team's knowledge is more skewed towards simulation and less on FET sensors. Hope you can build a more inter-disciplinary team :)

16.  Thank you

Author Response

(The authors gave the same response as above.)

Round 2

Reviewer 1 Report

It seems that the authors made substantial revisions according to my previous comments judging from the response letter. However for some reason, the revised manuscript downloaded from the system (v2) does not show any difference with the old version (v1). Therefore, it is difficult to evaluate the quality of the manuscript. Please double check whether the correct version of manuscript was uploaded.

Reviewer 2 Report

I checked the manuscript. The authors revised it according to all my previous revisions, so I believe it is now suitable for publication.

Round 3

Reviewer 1 Report

In the updated manuscript, the authors have addressed all my previous concerns. The work is publishable in the current form.